# Role of Metabolism and Metabolic Pathways in Prostate Cancer

**DOI:** 10.3390/metabo13020183

**Published:** 2023-01-25

**Authors:** Uddesh Ramesh Wanjari, Anirban Goutam Mukherjee, Abilash Valsala Gopalakrishnan, Reshma Murali, Abhijit Dey, Balachandar Vellingiri, Raja Ganesan

**Affiliations:** 1Department of Biomedical Sciences, School of Biosciences and Technology, Vellore Institute of Technology (VIT), Vellore 632014, India; 2Department of Life Sciences, Presidency University, Kolkata 700073, India; 3Stem Cell and Regenerative Medicine/Translational Research, Department of Zoology, School of Basic Sciences, Central University of Punjab (CUPB), Bathinda 151401, India; 4Institute for Liver and Digestive Diseases, College of Medicine, Hallym University, Chuncheon 24252, Republic of Korea

**Keywords:** PCa, MetS, metabolism, androgen, implication

## Abstract

Prostate cancer (PCa) is the common cause of death in men. The pathophysiological factors contributing to PCa are not well known. PCa cells gain a protective mechanism via abnormal lipid signaling and metabolism. PCa cells modify their metabolism in response to an excessive intake of nutrients to facilitate advancement. Metabolic syndrome (MetS) is inextricably linked to the carcinogenic progression of PCa, which heightens the severity of the disease. It is hypothesized that changes in the metabolism of the mitochondria contribute to the onset of PCa. The studies of particular alterations in the progress of PCa are best accomplished by examining the metabolome of prostate tissue. Due to the inconsistent findings written initially, additional epidemiological research is required to identify whether or not MetS is an aspect of PCa. There is a correlation between several risk factors and the progression of PCa, one of which is MetS. The metabolic symbiosis between PCa cells and the tumor milieu and how this type of crosstalk may aid in the development of PCa is portrayed in this work. This review focuses on in-depth analysis and evaluation of the metabolic changes that occur within PCa, and also aims to assess the effect of metabolic abnormalities on the aggressiveness status and metabolism of PCa.

## 1. Introduction

There is no universal definition for cancer, but it is defined as the uncontrol division of the cell that leads to invading the cell basement or metastasizing to other body organs [1,2,3]. Benign and malignant tumors can be differentiated based on their location. A benign tumor remains at the primary site. In contrast, malignant tumor cells invade the cell membrane and enter the blood vessel via intravasation [2]. 

The benign prostate tumor might limit glycolysis and favor high oxidative phosphorylation. In contrast, the increased glycolysis becomes an advanced castrate resistance PCa characteristic. Changes occur in prostate epithelial cells—a unique metabolic phenotype, during the movement of the tumor from prostatic intraepithelial neoplasia (PIN) to metastasis. In that case, normal prostate epithelial cells depend on glycolysis and use glucose to produce citrate. Simultaneously, PCa cells gradually re-activate mitochondrial phosphorylation with glucose metabolism, reducing citrate production. Androgen receptor (AR) in PCa supports metabolic and biosynthetic demands by reprogramming cellular metabolic pathways: mitochondrial respiration, aerobic glycolysis, and de novo lipogenesis [4,5,6,7]. Fatty acid (FA) synthesis may be an early event in prostate tumor formation and is also linked with disease progression [8,9]. Amino acid (AA) metabolism maintains the AA pool required in PCa progression. Glucose, lipids, and nitrogen precursor are purines and pyrimidines required for nucleic acid (NA) synthesis. AA synthesis and its metabolism in PCa focused on anaplerosis compared to energy production [10,11]. 

Several studies have focused on the characterization of PCa cell metabolic profile and pointed to the biological mechanism involved in disease progression. A study by Gómez-Cebrián et al., 2020, reported that healthy prostate cells show reduced citrate oxidation with lowering m-aconitase (ACO) activity due to the effect of high zinc concentration. It also slower TCA cycle metabolism [12]. Healthy prostate cells mostly rely on glycolytic pathways for energy [13,14]. The ultimate read-out of gene-environment interactions is obtained by measuring many low molecular weight substances (metabolites) in biofluids using metabolomics which may help identify novel risk factors for PCa. Several studies investigated the relationship between pre-diagnostic plasma, serum metabolite level, and PCa risk incidence [15,16,17,18].

## 2. A Link between PCa and MetS

MetS-like insulin resistance (IR), adipose tissue generated adipocytokines, and obesity are the factors that play a significant role in cancer development [19]. According to recent studies by Gacci et al. and Avgerinos et al., the mechanism and metabolic pathways are not fully characterized [20,21]. However, MetS and Cancer have some associations, including alteration in IGF-1 synthesis and signaling pathways due to IR [20], overproduction of sexual hormones [21,22], adipose tissue accumulation [20], fluctuations in sleep patterns, and abrupt dietary changes [21]. This factor leads to cancer cell proliferation and survival by alterations in an increase in certain events, such as proliferation and decreased apoptosis. This evidence is a considerable link between obesity and cancer caused by chronic inflammation, microbiome dysregulation, and IR [20]. MetS appear as a significant influencer in cancer cell progression. For example, PCa with MetS appears to increase cell death chances with high-grade tumor amplification [21]. 

MetS could be the potential risk factor for PCa development. However, previous studies reported an inconsistent link between PCa and MetS [23]. Simultaneously, some studies reported that MetS could increase PCa development [24,25]. In contrast, some studies reported no relation and negative association between MetS and PCa [26]. Recently, a study by Hammarsten et al. reported that MetS and its components might reduce the serum prostate-specific antigen (PSA) [27]. Gao et al. conducted the study in a Chinese cohort to explore the relationship between MetS and PCa. They found that the PCa incidence in men over 40 years of age was 0.1%. Based on this, it was concluded that MetS and obesity were not the PCa risk factor, whereas PSA levels with high age were the risk factor for PCa [28]. This relationship is controversial because, in Asian population studies, these inconsistencies have also been observed [29].

## 3. MetS and Incidence of PCa

PCa is the third most common malignancy and the second leading cause of death in the US [30]. According to data from 2020, it is estimated that there are 1.4 million diagnosed cases with an average age of 66 years and 360 thousand death/year with PCa. Additionally, countries such as the Caribbean, Australia, Europe, North America, and South Africa have a positive and negative relationship with PCa with high frequency [31,32,33,34]. It grows in an androgen-dependent state, and survival is based on androgen deprivation therapy (ADT)—the first line of treatment for advanced disease [35]. Studies also reported that MetS is linked with a high risk of PCa development and progression [36,37]. 

An international study has confirmed that in 32 out of 40 countries, the incidence of PCa is increasing, while in another eight countries is relatively stable [38,39]. According to a recent meta-analysis, MetS significantly increased the high-grade PCa incidence while having little impact on the prevalence of PCa [21]. Additionally, another study revealed that racial and geographic factors influence the association between MetS and PCa. MetS raises the risk of PCa in European countries, whereas it has little to no impact on PCa risk in the United States and other Asian countries [40]. A prospective population-based study conducted in Finland with 1880 males who had no prior history of diabetes or cancer at baseline and were followed for an average of 13.2 years stated that the individual factor of the MetS has been related to an elevated risk of PCa [41]. In a clinical trial, the patients with no prior history of PCa were screened to identify the link between PCa risk and MetS via transrectal ultrasound-guided prostate biopsy [42]. Lifestyle changes may alter the MetS-related disease progression. This may also treat or prevent the disease condition [43,44] (Table 1)**.**

## 4. MetS-Like Components on PCa Development

Obesity, diabetes, hypertension, and hypercholesterolemia are the MetS-like components [52]. Whereas variables such as elevated fasting glucose, triglycerides, and high blood pressure are not identical to the classic definition of MetS [53]. Hypertension and abdominal obesity are common in PCa patients, whereas diabetes is not linked with PCa risk. Beebe-Dimmer et al. reported that PCa-associated MetS in American-African men differs by race [24]. The study by Sourbeer et al. stated that more than three MetS-components are associated with high-grade PCa, not with an overall or low-grade PCa [53,54]. However, several studies about the relationship between MetS and PCa have been inconsistent [23]. Additional epidemiologic data are needed if Mets is a risk factor for PCa development [26]. Future studies must determine whether to prevent MetS and reduce the risk of aggressive PCa. 

Several biological mechanisms explain that MetS can lead to an increased risk of high-grade PCa, e.g., high cholesterol levels [55]. The altered level of IGF-1 [56], leptin [57], and adiponectin [58] in MetS conditions can also be linked with PCa risk. The pro-inflammatory state, including the increased C-reactive protein (CRP) level, interleukins-18 (IL-18), IL-1β, IL-6, and Tumor necrosis factor-α (TNF-α), is associated with MetS, which has been ultimately linked to PCa [59,60,61]. A MetS-linked component type II diabetes is related to low PCa risk due to pancreatic β-cells damage mediated hypoinsulinemia [62,63]. In contrast, hyperinsulinemia (HI) is associated with a high risk of PCa death [64]. These are the conditions making a clear association of MetS with PCa. There is a need for more extensive studies with multiethnic groups.

## 5. The Metabolic Phenotype of PCa

Metabolic phenotyping has developed into an effective method for discovering novel molecular biomarkers and metabolic vulnerabilities that may represent novel therapeutic options in cancer diseases [65,66]. The zinc (Zn) deposition in prostate cells leads to mitochondrial aconitase (ACO2) inhibition, lowering the citrate oxidation and consequently decreasing tricarboxylic acid (TCA) cycle metabolism [10]. In response to low Zn levels in PCa, reverse this condition [13]. With these conditions, it is reported that TCA cycle metabolism in PCa leads to decreased citrate levels and increased TCA cycle intermediates, including malate, fumarate, succinate ect [67,68,69]. The monocarboxylate transporters (MCT) expression levels differ PCa patients. MCT-producing phenotype related to aggressive PCa. Inhibiting its activity leads to a faster accumulation of toxic metabolic products [70]. The lactate shuttle expression in PCa can be a potential biomarker for the diagnosis and prognosis of the disease [71]. Glutaminase-1 performs a glutaminolysis, while its expression is upregulated in PCa. Blocking its action may dysregulate glutamine-based energy production in PCa [72,73]. Studies have shown that arginine maintains the malignant phenotype in PCa and is also needed for PCa growth, while the exact mechanism is not well understood [74]. PCa cells generate fatty acids by de novo lipid synthesis to obtain energy. This change to a lipid-producing phenotype marks a significant turning point in the development of PCa [13,75]. Warburg effect is characterized by alteration in preferential energy-producing pathways. The cancer cells follow aerobic glycolysis to produce ATP, while normal cells follow oxidative phosphorylation for ATP production [76,77].

## 6. Metabolic Regulation of PCa

The mechanisms involved in the transformation stages of PCa are not well understood. Metabolic reprogramming of cancer cells characterizes malignant changes, which can also become increasingly apparent. The molecular aspects of this reprogramming vary between cancer types and individual cells within a malignancy. Using glucose to generate citrate that is released as part of the seminal fluids is an unusual and inefficient energy metabolism for normal prostate epithelial cells. During transformation, PCa cells switch from an inefficient to efficient energy metabolism [3,78,79,80,81] (Figure 1).

### 6.1. Glycolysis

Positive [vascular endothelial growth factor [VEGF], IL-8, and stromal cell-derived factor [SDF]-1/(CXCL12)] and negative regulation of the angiogenic cascade are examples of hypoxia-inducible genes that are known to play an essential role in the initial stages of tumor adaptation to hypoxia. Hypoxia-inducible genes similarly regulate necessary glycolytic enzymes for anaerobic metabolism. These molecules are essential for producing growth-related macromolecules and fulfilling the high-energy needs of expanding tumors [82]. As a result, the so-called “Warburg effect,” in which cancer cells switch from producing energy via oxidative phosphorylation to doing so via glycolysis, is widely believed to be an essential feature of cancer cells and is linked to more rapid tumor growth [83,84,85]. 

One of the enzymes in the glycolytic process, phosphoglycerate kinase (PGK1), generates ATP. In PCa, PGK1 is frequently overexpressed and is controlled by hypoxia-inducible factor-1a (HIF-1a) [86,87]. PGK1 maps to a region of the X chromosome (Xq11–Xq13) [88] that has been linked to an increased risk of developing PCa, hypospadias, and androgen insensitivity in families. PGK1 plays a role in DNA replication and repair by acting on DNA in the nucleus. Surprisingly, tumors extracellularly release PGK1, which can act as a disulfide reductase. It has the potential activity to cleave plasminogen to yield angiostatin (a vascular inhibitor) [89,90,91,92,93,94]. Therefore, it is probable that overexpression of PGK1 will slow tumor growth by reducing angiogenesis. Due to this, tumor growth may depend on a finely tuned balancing act between the hypoxic response, which is required to produce pro-angiogenic factors and events necessary for anaerobic metabolism (such as PGK1). Controlling PGK1 secretion is a significant obstacle. According to recent investigations, adipocytes, which have been demonstrated to control metabolism in the primary tumor via CAFS, are possibly responsible for the transition to the Warburg effect in PCa metastatic cells. High glycolytic rates and increased HIF1 synthesis result from PCa cells co-culturing with adipocytes without oxygen. The accumulation of lactate and the inhibition of oxidative phosphorylation (OXPHOS) are caused by the transcription of the Warburg-associated genes driven by the HIF1 increase. However, PCa cells treated with CM from PCa/adipocyte co-culture undergo remarkable metabolic reprogramming. However, PCa CM-treated adipocytes release a substantial amount of free glycerol and show an increase in the lipolytic enzyme adipose triglyceride lipase (ATGL), indicating a PCa-induced lipolytic phenotype [7,95]. Despite the lack of experimental proof in this scenario, it is well-established that glycerol participates in the glycolytic process to provide energy for cancer cells [96,97,98]. Altogether, established results support the hypothesis that PCa cells tamper with their surroundings to produce metabolic intermediates utilized by cancer cells. Consequently, the tumor’s microenvironment provides metabolic support for the growth of the tumor, making it an optimal setting for the survival and proliferation of cancer cells [78,99].

Citrate can meet PCa cancer cells’ energy needs because citrate excretion is downregulated [100]. Schöpf et al., 2020 [101] found that glutamate and malate drove OXPHOS capacity in benign human prostate tissue, while in malignant tissue, succinate and pyruvate drove energy production and compensated for the diminished N-pathway capacity, correlating with results from PCa cell lines by Weber et al., 2018 [102]. Researchers Badder et al. (2019) [103] found that OXPHOS activity in PCa cell models revealed the significance of pyruvate in maintaining tumor growth. Recent in vitro research by Zadra et al., 2015 shows that modifying AMPK’s metabolic activity slows the development of PCa [104]. Inhibiting glutamine absorption reduces PCa cell proliferation and invasion, according to an in vitro study by Wang et al., 2015 [73].

### 6.2. Gluconeogenesis

It is generally known that many cells can create energy (ATP) from alternate fuels, such as ketone bodies or fatty acids, in functioning mitochondria, even when glucose is scarce. [105,106]. Since the gluconeogenic and glycolytic processes exchange intermediates, gluconeogenesis could be a backup source of biosynthetic precursors when glucose is scarce [107,108]. 

The availability of nutrients is a constant source of stress for cancer cells, and disrupting their adaptive responses could be a proper treatment strategy. Cancer cells have been found to generate critical metabolites using shortened gluconeogenesis versions by expressing phosphoenolpyruvate carboxykinase (PEPCK, PCK1, or PCK2). Gluconeogenesis is the metabolic route that branches off glycolysis and employs lactate or amino acids as substrates. There is evidence that PCK1 and PCK2 are essential for developing several malignancies. In contrast, the downstream gluconeogenesis enzyme fructose-1,6-bisphosphatase 1 (FBP1) suppresses glycolysis and tumor growth through non-enzymatic processes. It has been found that PCK2 expression in PCa metastases is higher than in normal prostate cells or primary tumors [109]. Overall survival was significantly lower in PCa patients with highly expressed PCK2. In this work, PCa tumor-initiating cells (TIC) expressed considerably more PCK2 than other PCa cell types. Silencing PCK2 decreased the number of TIC, reduced their sphere-forming ability, and stopped the formation of PCa nodules in mice. Silencing PCK2 decreased glycolysis, indicating that PCK2 plays a role in driving the switch to glycolysis. Notably, TIC survival was marginally improved by PCK2 silencing in a glucose-free media. Acetyl-CoA levels, measured by the acetylation of lysine residues in histones and total proteins, were increased after PCK2 silencing. It was owing to increased synthesis from citrate via ATP citrate lyase (ACLY) [109,110].

### 6.3. Lipid Metabolism in PCa 

Researchers have investigated the involvement of lipid metabolism in the development and progression of PCa [111]. It has a crucial role in the malignant phenotype expression [112,113]. During PCa development, PCa cells undergo adaptive metabolic alterations to maintain growth and progression [114]. The high expression and activity of cell membrane phospholipids biosynthesis enzyme known as choline kinase with other lipid molecules, including phosphatidylethanolamine, phosphatidylcholine, and glycerophosphocholine, are reported in PCa [115]. The α-methyl acyl-CoA racemase (AM-ACR) is the peroxisomal enzyme required to oxidize branch-chained fatty acids (FAs). AM-ACR is highly expressed in PCa cells and increases FAs oxidation as an energy source to support PCa progression and survival [116,117,118]. 

Huang et al. stated that sterol regulatory binding protein 1 (SREBP-1) increases the NADPH oxidase-5 and lipogenesis, which plays a significant role in PCa growth, and metastasis [119]. The lipogenic phenotype of PCa is associated with de novo lipogenesis, and cholesterogenesis is carried out by converting derived citrate from the TCA cycle and oxaloacetate to acetyl-Co-A by ACLY in the cytosol. Acetyl-CoA carboxylase (ACC) converts acetyl-CoA to malonyl-CoA and oxaloacetate into pyruvate, which can re-enter the mitochondria for further utilization. This leads to high activation of de novo lipogenesis and cholesterogenesis. Whereas it has been found that there is increased expression of ACC, ACLY, and FA synthase in PCa [120,121].

#### 6.3.1. Cholesterol Metabolism

The plasma membrane contains cholesterol, approximately one-third of lipids [122]. Cholesterol is crucial for cell proliferation, while its depletion leads to cell arrest [123]. It maintains cancer stem cells and activates oncogenic Hedgehog signaling [124,125]. Compared with normal cells, cholesterol is highly needed for PCa cell proliferation [126,127]. The elevated cholesterol level was transferred to high-density lipoprotein (HDL) particles by ATP-binding cassette transporter A1 (ABCA1) and G1 (ABCG1) [128]. Whereas the excessive intracellular cholesterol returns to the intestine or liver by HDL particles, where it is recycled. It is delivered to steroidogenic organs for hormone synthesis [129,130]. Acyl-Co acyltransferases esterify excessive free cholesterol and are stored in cytoplasmic lipid droplets to avoid cholesterol toxicity. They are exported from cells via ABCA1 and ABCG1 [128,131]. PCa can be easily diagnosed with high cholesterol levels due to elevated PSA. However, the studies on these did not consider [132]. Whereas highly expressed LDLR in PCa cannot uptake enough cholesterol from the blood. Therefore, the cholesterol level in the blood is the same before and after prostate surgery [133,134,135]. 

Cholesterol metabolism is associated with PCa pathogenesis and acts as a substrate for intratumoral androgen biosynthesis [136]. Castration-resistant PCa cells (CRPCa) are highly expressed CPY17A1 enzymes that de-novo synthesizes androgens [137]. It emphasizes the significance of steroid biosynthesis as a critical biological process connecting PCa and cholesterol [115,138]. The high-grade PCa with serum cholesterol level gives a positive correlation [139], several cholesterol metabolic regulator abnormalities found in in-vivo [140] and in-vitro cancer progression models [141,142], and statin lowering the risk of PCa occurrence, which also be used for a cholesterol-lowering therapy [143,144,145].

#### 6.3.2. Caveolin-1-Mediated Metabolism

Caveolae formation requires Polymerase-I and transcript release factor (PTRF) as CAV1 and CAVIN1. These are the isoform of Cav-1 found in Cav-α and β. Studies identified Cav-1 as a substrate for proto-oncogenic kinases ABL1 [146], FYN [147], and SRC [148]. It can be phosphorylated in response to rapamycin (mTOR) complex 2 (mTORC2) signaling [149] and epithelial growth factor (EGF) [150]. Caveolin-1 (Cav-1) is also known as a lipid chaperone. It is a transporter molecule that facilitates the mechanoprotection of cell membranes, cellular lipid homeostasis, and endocytosis and exocytosis [151]. Similarly, it modulates transmembrane signal transduction and microdomain arrangement in the membrane [152]. It plays a role in transporting chemokines, proteins, LDL, and HDL [153]. 

Cav-1 disturbs several metabolic pathways in PCa cells [154]. It has a dynamic role in cancer [155]. It is found in the activation and regulation of integrins and cadherins, receptor tyrosine kinases (RTK). G-protein coupled receptors (GPCR) [156]. High expression of Cav-1 is linked with aggressive tumor phenotypes [157] and metastasis potential [158]. It is also associated with radio drug and multidrug resistance [159]. Tumor-supportive oncometabolism studies stated that Cav-1 enables re-modification of cancer cell lipid metabolism toward increased sphingomyelins catabolism to ceramide derivatives and changes ceramide metabolism in cancer cells. These all lead to the efflux of Cav-1-sphingolipid particles containing mitochondrial proteins and lipids and increased glycosphingolipid synthesis [160].

### 6.4. Mitochondrial Metabolism in PCa

Mitochondrial is also known as “the powerhouse of the cell” and plays a major role in reprogramming cancer cell metabolism [161]. It is an important bioenergetic hub that maintains ATP production, ROS generation, calcium signaling, redox balance, and apoptotic pathways [162,163]. Almost all metabolic fuels combine with acetyl-Co-A and can be converted entirely into CO_2_, water, and ATP, directing it into catabolic processes. The major metabolic pathways in mitochondria involve the catabolism of biomolecules and energy production, such as the TCA cycle, the electron transport chain (ETC), FAO, and OXPHOS [164]. 

In PCa metabolism, glucose and aspartate produce citrate and are secreted in prostatic fluid. The normal prostate epithelial cells use glucose to maintain their physiological citrate secretion, which is oxidized in the TCA cycle in PCa. In contrast, pyruvate is the primary source in the TCA cycle of PCa cells. During the PCa cell transformations, PCa cells consumed citrate to enhance OXPHOS and used it to fuel lipogenesis [6]. Understanding how each of these mechanisms works in various cancer types may offer novel therapy ideas as they all have potential roles in the development and malignant phenotypes maintenance [165] (Figure 2).

Multiple investigations have shown that NF-κB aids PCa development and progression by encouraging cell survival, proliferation, and invasion [166,167,168]. PCa has been linked to multiple members of the NF-κB family, although p52 is particularly crucial. RelA, RelB, and c-Rel are all transactivating subunits [169,170,171,172,173]. In particular, Bcl-3 is required for the survival of PCa cells after chemotherapy [174]. It has been hypothesized that so-called cancer stem cells are essential for the emergence of resistance to treatment, and a population of AR-negative PCa stem cells with constitutive NF-κB activity was lately identified [175,176]. Castration or AR antagonist resistance is primarily attributable to NF-κB, and promising outcomes have been shown in combination treatments that target both NF-κB and AR antagonists [177,178,179]. Aspirin and its active metabolite salicylic acid (SA) has been shown to effectively suppress the growth of the androgen-independent PCa cell line DU-145, adding further evidence for the significance of NF-κB in PCa proliferation [180]. However, aspirin may affect PCa by inhibiting cyclooxygenase (COX), the enzyme responsible for prostaglandin formation. Prostaglandins are crucial for the proliferation of PCa cells [181,182,183]. SA’s direct effect on NF-κB signaling at several levels is the most likely explanation for free SA’s inhibitory actions, as free SA cannot inhibit COX [184]. High mobility group box protein 1 (HMGB1) overexpression is correlated with tumor cell proliferation and aggressiveness in PCa [185,186,187]. Long-term Aspirin or NSAIDS use has been shown to reduce the risk of PCa in epidemiological studies [187]. AR inhibits expression induced by canonical NF-κB (RelA/p50) but appears to increase activation of non-canonical NF-κB favorably [188]. Loss of androgen repression of NF-κB target genes is related to poor prognosis in metastatic PCa [189], suggesting that activation of non-canonical NF-κB may be a crucial stage in establishing androgen independence. In addition to inducing metabolic reprogramming of PCa cells via activation of genes for glucose absorption and metabolism [169,190], the p52 subunit can also stimulate AR signaling, which adds to androgen-independent proliferation [191,192]. But there may be yet another layer of crosstalk between NF-κB and AR, as the classical IKKα and IKKβ upstream of NF-κB may also effectively regulate AR activation through phosphorylation [193,194,195,196,197,198].

#### Glutamine Metabolism

Glutamine (Gln) is synthesized by the human body. It is an essential source for normal and cancer cell survival, as they will die under Gln-depletion conditions [199,200,201]. Under the catabolic condition, it is consumed by the gastrointestinal (GI) tract, kidney, and immune compartment. It is an energy source for biosynthesis and homeostasis [202,203,204]. An in-depth understanding is required for regulating Gln-metabolism in cancer cell development and proliferation associated with hormonal studies such as progesterone, thyroid hormone, androgen, estrogen, prostaglandin, and insulin [205]. 

Previously, Gln was found to maintain the redox state by reducing ROS generation and producing GSH in PCa [199,206]. Lipids, pyruvate, and succinate appear as the primary energy source and biosynthesis in PCa. Gln has less impact in low-risk primary PCa [136,137] and has a prominent role in advanced PCa [207]. The PCa with high MYC expression acquired oncogene-driven Gln addiction due to the high demand for Gln for cancer cell progression [200,208]. It plays a significant role in ATP production by participating in the TCA cycle [72,204,206]. Gln enters the mammalian cells through AA transporters such as SLC1A5/ASCT2 and undergoes deamination in the mitochondrial cell by the action of glutaminase-1 (GLS-1) (kidney type glutaminase) and GLS-2 (liver type glutaminase), and finally transformed into glutamate (Glu) [72,209]. Next, by the action of glutamate dehydrogenase (GLUD1 or GLUD2) on Glu, it is converted into α-ketoglutarate (α-KG). It participates in ATP, NADH, and FADH2 production by entering the Kreb cycle. Additionally, malate (a Kreb cycle intermediate) leaves the process. It produces pyruvate and NADPH [210], whereas oxaloacetate (OAA) is used for nucleotide synthesis by converting it to aspartate. Citrate undergoes cataplerosis and the production of acetyl-CoA and lipids [202,211]. Therefore, Gln-metabolism benefits PCa by enhancing tumor development.

### 6.5. Neuroendocrine PCa (NEPCa) Metabolism

NEPCa is a newly arisen, more often form of aggressive disease. Androgen deprivation leads to reduce AR signaling in NEPCa. The NEPCa are also distinguished for their high neuroendocrine lineage markers expression, which are synaptophysin, chromogranin-A, and enolase [10,212]. Compared to prostate adenocarcinoma, mutational genetic changes in Rb1 and Tp53 and MYCN and AURKA amplification are more dominant in NEPCa. At the same time, MYCN is involved in neuroendocrine lineage reprogramming. It increases histone acetylation with mitochondrial export of acetyl groups and DNA accessibility [213]. Functional loss of Rb1 and Tp53 enabled the pluripotency networks activations via expressional activation of SOX2 (a transcriptional factor), epigenetic modifier, and enhancer of zeste-homolog 2 (EZH2) [214,215]. In glycolysis, pyruvate is generated and converted to acetyl-CoA, which plays a central role in regulating histone acetyltransferase (HAT) enzyme activity. In treatment, the high expression of histone lysine demethylase KDM8 induces NEPCa tumor to reprogram metabolism towards aerobic glycolysis [216,217]. The enhanced glutamine uptake and increased glycolysis elevate the pyruvate and acetyl-CoA production in NEPCa [218]. The highly activated glycolysis associated with MCT-4 mediates lactic acid regulation. It is the most clinically relevant metabolic feature in NEPC [219].

## 7. Effect of Myokines in PCa

Myokines are the cytokines produced by skeletal muscle and travel through a circulatory system, such as growth factors, metallopeptidases, and several other factors [220,221,222]. These factors may have beneficiary effects on the liver, reducing insulin resistance (IR) and adiposity, the immune system, and improving glucose uptake [223]. Their levels keep on changing as IL-6 [224,225], IL-10, IL-15 [226], oncostatin M [227] and decorin [228], irisin [229], myostatin, and secreted protein acidic risk in cysteine (SPARC) [230,231] may suppressing cancer cell proliferation and inhibiting the epithermal to mesenchymal cells transition (EMT), also accelerating apoptosis via cell cycle arrest. During exercise, skeletal muscle produces secret fibroblast growth factor-21 (FGF21), which can protect against low-grade systemic inflammation and reduce T2D risk via upregulating glycogenesis, downregulating gluconeogenesis, and lipogenesis in the liver [223]. IR, HI, and hyperlipidemia are associated with obesity, generating a tumor-favorable environment (TFE). Exercise-induced myokines can influence TFE by controlling adipose tissue and adipocytes [232]. It suggests that myokines have a potential protective effect on cancer. However, limited evidence suggests a direct association with tumor suppression [231,233,234].

## 8. Role of Androgens in PCa 

In normal prostate cells, Zn accumulation inhibits m-aconitase, which blocks the citrate to isocitrate conversion in the TCA cycle. Androgen carries out this process and results in a disturbed TCA cycle with an elevated citrate level in prostatic fluid. Whereas Zn concentration is low in the PCa and cannot be accumulated for a long time. Due to this, the m-aconitase stays free and can regulate the TCA cycle to derive the energy and acetyl-CoA (anabolic substrate) for de novo lipogenesis [37,100]. 

Androgen drives the PCa cell proliferation and survival via the AR axis. It regulates cellular metabolism, including homeostasis, tissue differentiation, and lipid metabolism [235]. In addition, promoting expression and activation of the transcription factor is sterol regulatory element-binding proteins (SREBPs). SREBPs bind to fatty acids (FAs) in the gene promoter regions [236]. Androgen also has direct AR-binding sites in the promoter region of the FA synthase gene, and it may extend to many other lipogenic enzymes [237,238]. Androgens stimulate lipid metabolism enzymes’ expression and influence lipid profiles in PCa cells [120,239]. Researchers worked on elucidating androgen-mediated FA uptake and oxidation with modulating FA transporters in PCa proliferation [240,241]. De novo lipogenesis might aid PCa growth by supplying the raw materials needed for synthesizing membranes and signaling. PCa cells may prefer FAO to support their viability after tumors expand, their vascular network compromising access to oxygen and nutrients. Following dispersion, PCa cells may adapt their metabolism to favor aerobic glycolysis. At the same time, FAO and de novo lipogenesis remain highly active [95,242].

## 9. Conclusions

Cancer is the most common cause of death worldwide, so it is critical to develop innovative solutions to this issue. Humanity would have sensitive tools for detecting and effectively treating it in an ideal world. Therefore, fast and accurate diagnostic techniques are required for its detection [243,244,245].

We are learning more and more about the complex molecular mechanisms that underlie PCa metabolism. Characterizing and regulating PCa’s metabolism may be crucial for delivering individualized treatment for the disease. Preventing risk factors might be a valuable strategy in the urgent demand for novel medicines. MetS seem to have the opposite impact. Therefore, more research is required to determine the relationship between MetS and the development of PCa. Several metabolic changes occur during the development of a benign cell into a malignant cell. Several cellular cancer metabolism mechanisms are still poorly known, requiring a lot of research. Better diagnostic procedures and treatment options may be developed due to increased understanding.

## Figures and Tables

**Figure 1 metabolites-13-00183-f001:**
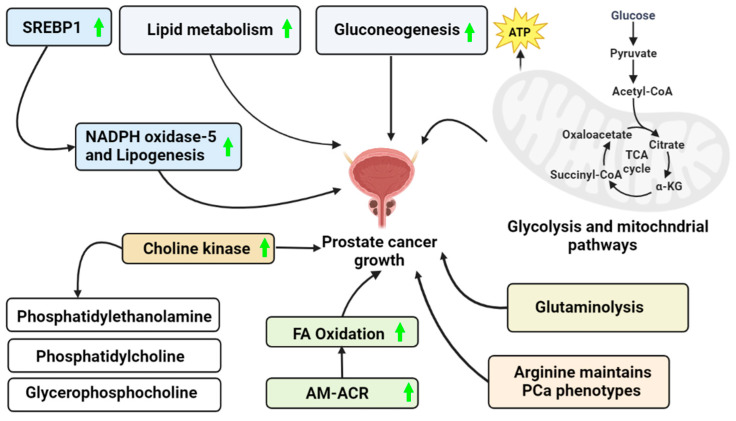
PCa growth via a different mechanism. This figure represents the possible ways to acquire the energy required to maintain PCa cell survival in an ATP-depleted cell environment. (Green arrow—Upregulation).

**Figure 2 metabolites-13-00183-f002:**
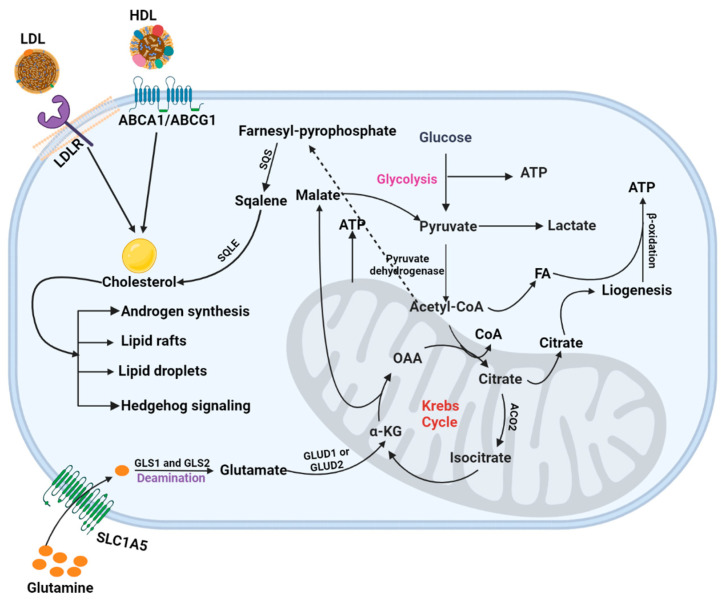
PCa metabolism. This figure illustrated the mechanistic pathways, including glycolysis, TCA cycle, and cholesterol and glutamine metabolism, involved in energy production in PCa cells.

**Table 1 metabolites-13-00183-t001:** MetS and PCa studies. This table summarizes epidemiological studies carried out on MetS and PCa relations.

Location	Study Population	Definition of Metabolic Syndrome Used	Inclusion Criteria	Exclusion Criteria	Study Design	Conclusion	Reference
Eastern Finland	1880 men	European Group for the Study of IR	Middle-aged (mean = 52.6 years); Obese men (BMI ≥ 27 kg/m^2^); lighter men	Cancer and diabetic men	Cohort population study	MetS increases PCa risk	[45]
Montreal, Canada	1937 men	NCEP-ATP III criteria	Age ≤ 75 years	Diabetic	Cohort	Inverse association between PCa risk and MetS	[46]
China	214 men	Chinese Diabetes Society criteria	Men with clinically localized PCa; January 2013–December 2015	Patients who received neoadjuvant hormonal therapy	Cohort	PCa Biochemical recurrence not associated with MetS	[47]
United States (US)	7082 men	Atherosclerosis Risk in Communities (ARIC) Study protocol	45–64 years	Women, cancer patients; not fasted for 8 h	Cohort	MetS marker of decreased risk of PCa	[48]
United Kingdom (UK)	220,622 men	NCEP ATP III	40–69 years; prior history of PCa testing; father with PCa	For men with other cancer diagnoses, PCa diagnoses within the first 3 years of follow-up	Cohort	No association between MetS and PCa risk	[49]
Caucasian	2322 men	NCEP and IDF	-	-	Cohort	NCEP-defined MetS is associated with PCa	[50]
Norway	29,364 men	NCEP ATP III	Age ≥ 20 years	Prevalent cancer- weight and height unspecified; unknown marital status	Cohort	PCa was not associated with several key MetS components	[51]

## Data Availability

Data are available from the authors on request (A.V.G.) due to privacy or ethical restrictions.

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
