# Peer review of "Role of Metabolism and Metabolic Pathways in Prostate Cancer"

_metabolites, 2023, doi:10.3390/metabo13020183_

Round 1
Reviewer 1 Report
Dear authors
The manuscript is suitable for publication and minor changes are required in terms of English writing and more figures in the field.
with best regards
Author Response
Reviewer 1:
Dear authors
The manuscript is suitable for publication and minor changes are required in terms of English writing and more figures in the field.
with best regards
Response: All the authors are thankful to the reviewer for evaluating the quality of the manuscript and recommending it for publication. We thoroughly checked the manuscript and corrected all the grammatical errors throughout the manuscript. The manuscript was edited for the proper English language, grammar, punctuation, and spelling by one or more of the highly qualified native English editors.
Reviewer 2 Report
1.It is suggested to add new in vitro and in vivo studies.
2.what is the suggestion of this study for future works?
3.Please discuss and compare your results with previous works and add suggestions for diagnosis using these metabolites.
4.It will be better to add the role of mitochondria and NFkB-mediated apoptosis.
5.Please add details for time period and dose selection from previous works.
6.More references for the discussion part of manuscript and bold your study novelty should be added: e.g.,
-DOI:10.1016/j.chemosphere.2022.134826
-DOI: 10.1016/j.trac.2019.05.004
-DOI: 10.1016/j.biopha.2018.01.117
Author Response
Reviewer 2:
1.It is suggested to add new in vitro and in vivo studies.
Response 1: We have added the some of latest studies in the section 6. 1 Glycolysis in the main manuscript. (Line number 205-214).
2.what is the suggestion of this study for future works?
Response 2: This review sheds light on different metabolic pathways which play a leading role in prostate cancer for their survival. This will help in understanding the in-depth mechanism and can be helpful in finding better therapeutic targets to overcome this disease. The futural direction and importance of this work is mentioned in the conclusion section as
Characterizing and regulating PCa's metabolism may be crucial for delivering individualized treatment for the disease. Preventing risk factors might be a valuable strategy in the urgent demand for novel medicines. MetS seem to have the opposite impact. Therefore, more research is required to determine the relationship between MetS and the development of PCa.
3.Please discuss and compare your results with previous works and add suggestions for diagnosis using these metabolites.
Response 3: Authors are thankful to the reviewer for their valuable suggestion. However, this comment is not applicable for this review, as our review mainly aims on ‘’Role of Metabolism and Metabolic Pathways in Prostate Cancer’’. we have already summarised the some of previously published article on this topic as follows
[7] A. P. Sousa, R. Costa, M. G. Alves, R. Soares, P. Baylina, and R. Fernandes, "The Impact of Metabolic Syndrome and Type 2 Diabetes Mellitus on Prostate Cancer," (in eng), Front Cell Dev Biol, vol. 10, p. 843458, 2022.
[10] P. Chetta and G. Zadra, "Metabolic reprogramming as an emerging mechanism of resistance to endocrine therapies in prostate cancer," (in eng), Cancer Drug Resist, vol. 4, no. 1, pp. 143-162, 2021
[16] F. Ahmad, M. K. Cherukuri, and P. L. Choyke, "Metabolic reprogramming in prostate cancer," (in eng), Br J Cancer, vol. 125, no. 9, pp. 1185-1196, Oct 2021.
4.It will be better to add the role of mitochondria and NFkB-mediated apoptosis.
Response 4: We have added the required data in section 6.4 Mitochondrial Metabolism in PCa in the main manuscript. (Line number 326-354).
5.Please add details for time period and dose selection from previous works.
Response 5: Authors are thankful to the reviewer for their valuable suggestion. However, this comment is not applicable for this review, as our review mainly aims on ‘’Role of Metabolism and Metabolic Pathways in Prostate Cancer’’.
6.More references for the discussion part of manuscript and bold your study novelty should be added: e.g.,
-DOI:10.1016/j.chemosphere.2022.134826
-DOI: 10.1016/j.trac.2019.05.004
-DOI: 10.1016/j.biopha.2018.01.117
Response 6: We have satisfied the suggested comment for improving the quality of the manuscript. The changes are made in track changed file.
Cancer is the most common cause of death in the world, so it's critical to develop innovative solutions to this issue. Humanity would have sensitive tools for detecting and effectively treating it in an ideal world. Therefore, fast and accurate diagnostic techniques are required for its detection [247-249].
The manuscript was edited for the proper English language, grammar, punctuation, and spelling by one or more of the highly qualified native English editors.
Reviewer 3 Report
This review, “Role of Metabolism and Metabolic Pathways in Prostate Cancer”, aimed to summarize the link between metabolic pathways and PCa. Overall, the review is comprehensive and will provide helpful insights to the field.
The following comments or suggestions, if can be addressed, would further strengthen this manuscript.
- In the introduction section, authors should 1) summarize previous reviews on this topic, if any; 2) Identify the main aim/purpose of the review, i.e., what’s the key message authors want to get across; 3) include a brief summary of the reviewing method used here; 4) include a brief summary of the structure of the review.
- Some sentences are grammatically incorrect, e.g., line 50, line 96, Line 167, etc. Authors should check and proofread the manuscript to eliminate these errors.
- The authors refer to a number of epidemiological studies that provide inconsistent evidence linking MetS and PCa. A table summarizing these epidemiological studies may be helpful to readers.
- There are some redundant sentences in each section. For example, line 96-99 should be moved to the introduction. Also, since sections 2-4 are all about epidemiological studies that link MetS and PCa (incidence and development), perhaps they should be merged into one section or sub-sections?
- Figure 2 and Section 6.4 illustrate the mitochondrial metabolic pathway, but the authors do not explain/review its specific relationship to PCa. Since this is a review article, the authors should summarize previous papers that have studied the mechanisms of mitochondria and PCa.
Author Response
Reviewer 3:
This review, “Role of Metabolism and Metabolic Pathways in Prostate Cancer”, aimed to summarize the link between metabolic pathways and PCa. Overall, the review is comprehensive and will provide helpful insights to the field.
The following comments or suggestions, if can be addressed, would further strengthen this manuscript.
Response: Authors are thankful to the reviewer for evaluating the quality of the manuscript, and for potential comments for improving its quality. We have addressed all the comments.
- In the introduction section, authors should 1) summarize previous reviews on this topic, if any; 2) Identify the main aim/purpose of the review, i.e., what’s the key message authors want to get across; 3) include a brief summary of the reviewing method used here; 4) include a brief summary of the structure of the review.
Response 1:
1) we have already summarised the some of previously published article on this topic as follows
[7] A. P. Sousa, R. Costa, M. G. Alves, R. Soares, P. Baylina, and R. Fernandes, "The Impact of Metabolic Syndrome and Type 2 Diabetes Mellitus on Prostate Cancer," (in eng), Front Cell Dev Biol, vol. 10, p. 843458, 2022.
[10] P. Chetta and G. Zadra, "Metabolic reprogramming as an emerging mechanism of resistance to endocrine therapies in prostate cancer," (in eng), Cancer Drug Resist, vol. 4, no. 1, pp. 143-162, 2021
[16] F. Ahmad, M. K. Cherukuri, and P. L. Choyke, "Metabolic reprogramming in prostate cancer," (in eng), Br J Cancer, vol. 125, no. 9, pp. 1185-1196, Oct 2021.
2) This review sheds light on different metabolic pathways which play a leading role in prostate cancer for their survival. This will help in understanding the in-depth mechanism and can be helpful in finding better therapeutic targets to overcome this disease. The futural direction and importance of this work is mentioned in the conclusion section as
Characterizing and regulating PCa's metabolism may be crucial for delivering individualized treatment for the disease. Preventing risk factors might be a valuable strategy in the urgent demand for novel medicines. MetS seem to have the opposite impact. Therefore, more research is required to determine the relationship between MetS and the development of PCa.
3) We have referred to the updated published manuscripts to understand the basic concept and presented it in a proper manner, which will help other researchers to find out new therapeutic targets or new therapeutic ways to treat prostate cancer.
4) The brief summary of the review has been given in abstract form in the main manuscript as
Prostate cancer (PCa) is the common cause of death in men. The pathophysiological factors contributing to PCa are not well known. PCa cells gain a protective mechanism via abnormal lipid signaling and metabolism. PCa cells modify their metabolism in response to an excessive intake of nutrients to facilitate advancement. Metabolic syndrome (MetS) is inextricably linked to the carcinogenic progression of PCa, which heightens the severity of the disease. It is hypothesized that changes in the metabolism of the mitochondria contribute to the onset of PCa. The studies of particular alterations in the progress of PCa are best accomplished by examining the metabolome of prostate tissue. Due to the inconsistent findings written initially, additional epidemiological research is required to identify whether or not MetS is an aspect of PCa. There is a correlation between several risk factors and the progression of PCa, one of which is MetS. The metabolic symbiosis between PCa cells and the tumor milieu and how this type of crosstalk may aid in the development of PCa is portrayed in this work. This review focuses on in-depth analysis and evaluation of the metabolic changes that occur within PCa, and also aims to assess the effect of metabolic abnormalities on the aggressiveness status and metabolism of PCa.
- Some sentences are grammatically incorrect, e.g., line 50, line 96, Line 167, etc. Authors should check and proofread the manuscript to eliminate these errors.
Response 2: We have removed the suggested lines to prevent any confusion in the main manuscript. The manuscript was edited for the proper English language, grammar, punctuation, and spelling by one or more of the highly qualified native English editors.
- The authors refer to a number of epidemiological studies that provide inconsistent evidence linking MetS and PCa. A table summarizing these epidemiological studies may be helpful to readers.
Response 3: We have added the Table summarizing epidemiological studies done on MetS and PCa relation. The table placed after the section 3. MetS and incidence of PCa in the main manuscript. (Line Number 109-111)
- There are some redundant sentences in each section. For example, line 96-99 should be moved to the introduction. Also, since sections 2-4 are all about epidemiological studies that link MetS and PCa (incidence and development), perhaps they should be merged into one section or sub-sections?
Response 4: We have re-positioned the epidemiological data from introduction section to section 3. MetS and incidence of PCa in the main manuscript.
- Figure 2 and Section 6.4 illustrate the mitochondrial metabolic pathway, but the authors do not explain/review its specific relationship to PCa. Since this is a review article, the authors should summarize previous papers that have studied the mechanisms of mitochondria and PCa.
Response 5: We have added the required data and tried to satisfy the suggested comment in 6.4 section. (Line Number 313-320)